# Association of Different ABO and Rh Blood Groups with the Erythroprotective Effect of Extracts from *Navicula incerta* and Their Anti-Inflammatory and Antiproliferative Properties

**DOI:** 10.3390/metabo12121203

**Published:** 2022-12-01

**Authors:** Saúl Ruiz-Cruz, Ricardo Iván González-Vega, Ramón Enrique Robles-Zepeda, Aline Reyes-Díaz, José Antonio López-Elías, Maritza Lizeth Álvarez-Ainza, Francisco Javier Cinco-Moroyoqui, Ramón Alfonso Moreno-Corral, Francisco Javier Wong-Corral, Jesús Borboa-Flores, Yaeel Isbeth Cornejo-Ramírez, Carmen Lizette Del-Toro-Sánchez

**Affiliations:** 1Department of Research and Postgraduate in Food, University of Sonora, Blvd Luis Encinas y Rosales S/N, Col. Centro, Hermosillo 83000, Mexico; 2Department of Medical and Life Sciences, Cienega University Center (CUCIÉNEGA), University of Guadalajara, Av. Universidad 1115, Lindavista, Ocotlán 47820, Mexico; 3Nursing Department, State University of Sonora, Av. Niños Héroes, San Javier, Magdalena de Kino 84160, Mexico

**Keywords:** red blood cells, erythroprotective effect, pigment-rich extracts

## Abstract

Previous studies have reported that different blood groups are associated with the risk of chronic degenerative diseases that mainly involve inflammation and neoplastic processes. We investigate the relationship between blood groups and the erythroprotective effect of extracts from *Navicula incerta* against oxidative damage as a proposal to develop drugs designed for people with a specific blood type related to chronic pathology. The study was carried out through the elucidation of the erythroprotective potential, anti-inflammatory and antiproliferative activity of *Navicula incerta*. Research suggests that the presence or absence of certain blood groups increases or decreases the abilities of certain phytochemicals to inhibit oxidative stress, which is related to the systemic inflammatory response involved in the development of different types of cancer. The pigment-rich extracts from *Navicula incerta* inhibit ROO•- induced oxidative stress in human erythrocytes on the A RhD+ve antigen without compromising the structure of the cell membrane. This result is very important, since the A antigen is related to the susceptibility of contracting prostate cancer. Similarly, it was possible to inhibit the proliferation of cervical (HeLa) and prostate (PC-3) carcinoma. The combinatorial analysis of different biological activities can help design phytochemicals as new candidates for preventive drugs treating the chronic degenerative diseases associated with a specific blood group.

## 1. Introduction

The polymorphic genetic antigen receptors are part of a blood group system that is made up of three alleles (A, B, and O) and four phenotypes (O, A, B, and AB) that are characterized by carbohydrate epitopes on the erythrocyte membrane surface [1]. The Rhesus (Rh) factor is a group of antigens that coat the surface of red blood cells (RBC), and the antigen D is the most important for transfusion medicine. An association between ABO and Rh blood groups and different types of diseases has been observed [2,3,4]. In this context, the effectiveness of the bioactive compounds in the different studies that support an association between the expression of the ABO and Rh groups with the risk of chronic degenerative diseases could depend on these groups’ expressions on the surface of RBC [3,4,5]. For example, blood group A has been associated with the incidence of different cancers and certain viral transmissions, such as COVID-19, while the O blood type was associated with duodenal ulcers and gastric carcinoma and the B type with pancreatic cancer. Thus, the search for natural bioactive compounds is currently the subject of several investigations [6,7,8,9].

These studies mainly involve inflammation and the neoplastic processes that begin when an inflammatory response occurs without injury [10]. Many of these studies have used erythrocytes as a cellular membrane model because it is similar to the lysosomal membrane. Stabilizing the lysosomal membrane is relevant to limiting the inflammatory response, since it inhibits the release of protease-type enzymes [11]. These enzymes degrade the connective tissue proteins that lead to the inflammatory responses involved with rheumatoid arthritis. Therefore, inhibiting the degradation of the proteins caused by the enzymes released by neutrophils is a well-founded reason to look for anti-inflammatory compounds that can act as enzymatic inhibitors [12]. However, anti-inflammatory drugs such as diclofenac sodium present adverse health effects, causing hepatotoxicity and gastrointestinal problems. Therefore, it is crucial to find novel and natural anti-inflammatory compounds.

To date, medicinal plants and macro and microalgae have been the main sources of phytochemicals with therapeutic applications in the pharmaceutical industry [13]. Bioactive compounds such as isoprenoids and carotenoids, among others, are present in traditional medicinal plants, exerting beneficial effects on health [14,15]. It is important that the bioactive compounds used to develop products that promote health do not exert a cytotoxic action on the blood cell lines. Consequently, they require blood biocompatibility tests. Currently, marine microalgae act as a key, providing therapeutic compounds without side effects for the development of drugs and nutraceuticals for use against chronic degenerative diseases [4]. Recently, substantial evidence in the literature has documented the potential use of marine microalgae as antioxidant [16], anti-inflammatory [17,18], antiproliferative [19], and erythroprotective [20] agents. Oxidative damage can be neutralized by antioxidants, which inhibit the oxidation and degradation of plasma membranes, thereby preventing the development of neurological [21], neoplastic [6,7,8,9], and hemolytic diseases [20]. Oxidative damage to erythrocytes alters their physical state and lipid asymmetry, manifesting structural, morphological, and functional changes, alterations present in different pathologies associated with cellular stress [22].

The diatom *Navicula incerta*, a marine microalga, is currently under pharmacological study. In our laboratory, the pigment-rich extract from this microalga has shown promising results in preventing the peroxyl radicals (ROO•) that induced hemolysis in human erythrocytes. However, there is evidence that the biological activity of *Navicula incerta* pigments is scant [20]. Several studies derived from microalgae have reported that carotenoids like fucoxanthin, fucoxanthinol, and neoxanthin show potent antiproliferative activity in their involvement in the mechanism of the cell cycle, apoptosis, and metastasis [19]. Furthermore, astaxanthin and β-carotene have been shown to inhibit the protein denaturation mechanisms involved in rheumatoid arthritis [17,18]. These pigments exhibit a high antioxidant capacity and erythroprotective effect [20]. However, the association of different ABO and Rh blood groups with erythroprotective effects has not been studied. These studies could enhance the use and development of functional foods, specifically preventing chronic diseases that affect a more susceptible population group. Additionally, studying their anti-inflammatory and antiproliferative properties will indicate the therapeutic potential of the biomolecules of marine diatoms. Therefore, this study aimed to determine the association of the different ABO and Rh blood groups with the antihemolytic activity of extracts from marine diatom *N. incerta* and their anti-inflammatory and antiproliferative properties.

## 2. Materials and Methods

### 2.1. Reagents

Reagents such as porcine pancreatic elastase (PPE, type IV), succinyl-Ala-Ala-Ala-p-nitroanilide (ESIV, elastase substrate IV), AAPH [2,2′-Azobis(2-methylpropionamidine) dihydrochloride], HPLC-solvents, bovine serum albumin (BSA), Tris-HCl biological buffer, and Triton X-100 were purchased from Sigma-Aldrich Co. (San Luis, MO, USA). All other chemicals and solvents were of the highest commercial grade.

### 2.2. Biological Material

The benthic diatom *Navicula incerta* was generously provided by the Centro de Investigación Científica y de Educación Superior de Ensenada (CICESE), Ensenada, Baja California, Mexico. The diatom was cultivated under laboratory-controlled conditions in 1 L flasks at a rate of 700 mL of culture. To maintain the culture, which had a pH of 7.0, homogeneous constant aeration and constant light (24 h) at 25 °C were applied. White wavelengths (400–750 nm) at 100 μmol photon/m^2^/s were used to maintain constant light, and they were produced electronically by Light Emitting Diode lamps (LED), which were regulated to achieve the desired intensity. A quantic spherical sensor Li-Cor 193SA was used to measure irradiance and wavelength. The culture conditions used in this study were taken from a previously optimized culture with the following conditions: F/4 medium with 0.44 mol·L^−1^ of nitrogen, 40 practical salinity unity (PSU, [g/kg]), and an age of culture of 3.5 days (in the logarithmic grown phase). For subsequent tests, the entire microalgal culture was harvested and washed with sodium formate to remove the salt. The total biomass collected was lyophilized (Yamato Scientific CO., LTD. Minami, Japan). The Red Blood Cells (RBC) with different blood groups (ABO and Rh) were donated by the Clinical Analysis Laboratory of the University of Sonora using a completely sterile vial and mixed with anticoagulant (EDTA) to prevent blood clotting. Blood samples were processed for analysis immediately after extraction. These samples contained approximately 4.7 to 6.1 × 106 cells/μL. The cell lines used in the study were purchased from the American Type Culture Collection (ATCC, Rockville, MD, USA). The cell lines were A549 (lung carcinoma), HeLa (cervical carcinoma), PC-3 (prostate carcinoma), LS-180 (colorectal carcinoma), and ARPE-19 (healthy retinal cell). They were cultured in Dulbecco’s modified Eagle’s medium (DMEM) and supplemented with 5% heat-inactivated fetal bovine serum (FBS). Antibiotics (2% penicillin-streptomycin) were included to avoid medium contamination. The cells were incubated (Fisher Scientific, Waltham, MA, USA) at 37 °C in a 5% CO_2_ atmosphere with 80–90% relative humidity. Under these conditions, they were reactivated for 24 h to carry out their doubling time. Cells lines were seeded in 25 cm^2^ plates at a 0.5 × 106 cells/well concentration. Later, when the cells reached confluence, they were subcultured by trypsinization (0.25% trypsin/0.03% EDTA) to break down the extracellular matrix and added to a 96-well cell culture microplate (Multiskan Go, Thermo Scientific, Waltham, MA, USA).

### 2.3. Extraction of Pigments from N. incerta

Three extracts of the pigments were obtained from 0.5 g of lyophilized microalgae, adding 50 mL of acetone (99% *v*/*v*), methanol (99% *v*/*v*), and ethanol (96% *v*/*v*) as solvent extractants. The extraction was ultrasonic-pulses-assisted (Generator ultrasonic pulses Branson Digital Sonifier Qsonica, LLC. E.U.A). The sample was subjected to 400 w of ultrasonic power for 3 pulses, at 500 mHz and 15 s per pulse at an amplitude of 30%. Then, it was kept in constant agitation for 1 h in darkness. The sample was centrifuged (Beckman model J2-21) for 15 min at 4 °C and 10,000× *g*. Later, each extract was concentrated in a rotary evaporator (Laborota 4000 Heidolph, Schwabach, Germany) at 45–50 °C. The supernatant was recovered for further analysis. The extracts were re-suspended in three different solvents, acetone (90%), methanol (99%), and ethanol (99%), where the concentrations of the extracts varied according to the biological activities tested [20].

### 2.4. Acute Toxicity Bioassay on Artemia Salina

The Artemia salina bioassay evaluated the acute toxicity of the extracts from *N. incerta* using the Molina-Salinas [23] and Leos-Rivas [24] method with some modifications. The toxicity of samples was classified according to Robles-García et al. [25], as indicated in Table 1. To hatch A. saline cysts (100 mg), it was necessary to use 1 L of sterile seawater (35 PSU). Artificial illumination (white light) and aeration (fish tank air pump) were constant for 48 h at 25 °C. Later, 120 μL with approximately 10–13 nauplii were taken from the flask and were added to a 96-well microplate. Additionally, 150 μL of each sample was added at different concentrations (5–1500 μg·mL^−1^). The initial number of nauplii (TV) was recorded. After 24 h of nauplii contact with the samples, the number of dead nauplii (TM) was quantified. The lethality percentage of each concentration was calculated using the following Equation (1):(1)$$\%\;\mathrm{of}\;\mathrm{lethality}=\frac{\mathrm{TM}}{\mathrm{TV}}\times 100$$

### 2.5. LC_50_ Determination

The mean lethal concentration (LC_50_) was determined by 24-h counts (concentration of the sample which kills 50% of the test organism after 24 h exposure) using the method described by Robles-García et al. [25]. The *A. salina* larvae were considered dead when they did not exhibit any movement during 10 s of observation.

### 2.6. Blood Biocompatibility Assay of N. incerta Extracts on ABO and Rh Blood Types

The hemolysis assay confirmed the blood biocompatibility of the *N. incerta* extracts with ABO and Rh blood types. This direct hemolytic assay was performed as described by Belokoneva et al. [26], with slight modifications. RBC samples were collected by venipuncture using EDTA tubes. A suspension of 10% erythrocytes was achieved by washing with phosphate-buffered saline (PBS) (0.15 M, pH 7.4) until the total plasma was removed (clear supernatant). Later, 150 μL of erythrocytes were mixed with 150 μL of extract (*v*/*v*) at a concentration of 150 μg·mL^−1^. The same procedure was carried out for each solvent control instead sample. The RBC suspension was incubated at room temperature and monitored for 6 h. After incubation, 1 mL of PBS was added, the RBC suspension was centrifuged at 2000× *g* for 5 min, and supernatants were recovered. The supernatant (150 μL) was measured at 540 nm in a 96-well microplate reader per triplicate. Triton X-100 1% and PBS buffer were used as positive and negative controls, respectively, under the same conditions as the samples. Different blood groups (ABO) and Rh (RhD+ve and RhD-ve) were evaluated for this study. The results were expressed as a hemolysis percentage, which was calculated using the following Equation (2):(2)$$\%\;of\;hemolysis=\frac{A_{sample}-A_{PBS}}{A_{Triton}-A_{PBS}}\times 100$$

### 2.7. Evaluation of the Erythroprotective Effect on Different ABO and Rh Blood Types of Groups

The erythroprotective effect was determined by inhibition hemolysis assay as described by Hernández-Ruiz et al. [27] with some modifications. The hemolysis of human erythrocytes was induced by a free-radical initiator, AAPH (2,2′-azobis-[2-methylpropionamidine]). An erythrocyte suspension (2%) was prepared to separate the plasma by centrifugation at 2000× *g* for 10 min at 4 °C (Thermo Biofuge Stratos, Thermo Fisher Scientific Inc., Waltham, Massachusetts, USA), by performing three washes with PBS (0.15 M) at pH 7.4. Then, in a test tube, 100 μL of blood suspension, 100 μL of extract (acetonic, methanolic, and ethanolic extract), and 100 μL of AAPH (40 mM) were combined. The same procedure was carried out for each solvent control instead sample. An aliquot of 100 μL of blood suspension, 100 μL of PBS, and 100 μL of AAPH (40 mM) were used as the hemolysis control, while 100 μL of blood suspension and 200 μL of PBS were used as the control without hemolysis. The mixture was incubated for 3 h at 37 °C with continuous shaking (30 rpm). After incubation, the mixture was diluted with 1 mL of PBS and centrifuged at 2000× *g* for 10 min at 4 °C. The supernatant was read at 540 nm on a 96-well microplate (Multiskan Go, Thermo Scientific, Waltham, MA, USA). The concentration of all the extracts tested was 333 μg·mL^−1^. Different blood groups (ABO) and Rh (RhD+ve and RhD-ve) were evaluated for this study. The results were performed as a percentage of hemolysis inhibition (% HI), and the value was calculated as the following Equation (3):(3)$$\%\;HI=\left(\frac{\mathrm{AAPH}1-\mathrm{HS}}{\mathrm{AAPH}1}\right)\times 100$$
where AAPH_1_ = optical density of the hemolysis caused per AAPH. % *HI* = percentage of hemolysis inhibition. HS = optical density of the hemolysis inhibition by each treatment.

The statistical association of ABO and Rh blood groups with the antihemolytic activity was determined by linear and logistic regression. This method allows estimation of the probability of a binary qualitative variable based on a quantitative variable. The statistical measure called Odds ratio (OR), which is a probability ratio, was used. These regression models are modeled by the logarithm of the probability of belonging to a group. The results are presented with values ranging from zero to infinity (0 to ∞) at a 95% confidence interval. Values less than 1 (OR < 1) are classified as the absence of an association between variables, while values greater than 1 (OR > 1) indicate an association between variables. 

After that, membranes of the different blood types of groups ABO and Rh were used to evaluate the morphological changes by optical microscopy (100× Eclipse FN1 microscope). This evaluation was performed immediately after reading the supernatant samples to the inhibition hemolysis test. To observe the RBCs’ cellular membrane changes, 50 μL of fresh plasma were added to the globular package. A drop of the above suspension was spread, creating a thin RBC layer over the slide. After this, the right staining method was necessary to visualize the RBC under the microscope. The micrograph results were compared with erythrocytes without AAPH and erythrocytes with AAPH. The micrographs are presented with a 100× magnification, and 5 µm scale bars are added to the micrographs to compare the RBC size. To capture images, the software NIS-Elements F was used. 

### 2.8. In Vitro Anti-Inflammatory Activity Screening

#### 2.8.1. Inhibition of Enzyme Porcine Pancreatic Elastase (PPE) Activity

The anti-inflammatory activity of the *N. incerta* extract was evaluated on the inhibition of the Porcine Pancreatic Elastase (PPE, Sigma, Type IV) enzyme, using the Lee et al. [28] method with some modifications. The PPE enzyme hydrolyses the substrate N-succ-(ala)-3-p-nitroaniline, releasing p-nitroaniline. This release was monitored for 60 min at 28 °C. The concentration used for each extract in the reaction system was 66.66 μg·mL^−1^. The substrate was prepared at a concentration of 1.015 mM in a 0.1 M solution of a Tris-HCl biological buffer with a pH of 8. The PPE was dissolved in 0.2 M of Tris-HCl (pH 8) at a concentration of 1.376 U/mL. The reaction was carried out in 96-well plates by adding 10 μL of extract, 40 μL of the enzyme PPE, and 100 μL of the substrate for a final volume of 150 μL. The reaction begins with the contact of the enzyme with the substrate, and the absorbance was recorded at 410 nm in a 96-well microplate reader. Diclofenac sodium (DS) was used as an anti-inflammatory control. The experiment was performed in triplicate. The results of the anti-inflammatory activity were expressed according to the following Equation (4):(4)$$\%\;\mathrm{of}\;\mathrm{inhibition}=\left[1-\frac{B}{A}\right]\times 100$$

*A* is expressed in μg·mL^−1^ of p-nitroaniline released without the inhibitor and *B* in µg·mL^−1^ of p-nitroaniline released in the presence of the inhibitor.

#### 2.8.2. Inhibition of Albumin Denaturation Assay (Antiarthritic Activity)

This assay was used to evaluate antiarthritic activity according to the methodology described by Govindapa [12] and Chandra et al. [29] with minor modifications. The reaction mixture consisted of 1 mL of 1% bovine serum albumin (BSA) and 200 μL of extract (200 μg·mL^−1^). The reaction mixture was adjusted at pH 6.8 using 1 N HCl. Later, it was incubated at room temperature for 20 min. After that, it was heated to 55 °C for 20 min in a water bath. After cooling at room temperature, the turbidity caused by albumin denaturation was measured in a 96-well microplate at 660 nm with 300 μL per well. A BSA mixture with a physiological solution was used as a control. Additionally, DS (200 μg·mL^−1^) was used as a control anti-inflammatory drug. The inhibition of albumin denaturation was calculated using the following Equation (5):(5)$$\%\;\mathrm{of}\;\mathrm{inhibition}=\frac{A_{control}-A_{Sample}}{A_{Control}}\times 100$$

#### 2.8.3. Heat-Induced Hemolysis and Hypotonicity-Induced Hemolysis Assays

Heat-induced hemolysis and hypotonicity-induced hemolysis assays were performed using the RBC membrane as a lysosomal membrane model to assess its stability. The analysis of RBC morphological changes by damage-induced heat and hypotonicity were conducted as explained in Section 2.6. Heat-induced hemolysis was carried out according to the methodology described by Agarwal et al. [22] with some modifications. The RBC were centrifuged at 2000× *g* for 10 min and washed three times with isotonic solution. Then, a 10% RBC suspension was prepared with PBS (0.15 M) at pH 7.4. An amount of 150 μL of 10% RBC suspension was mixed with 150 μL of each sample (300 μg·mL^−1^). The mixture was incubated in a water bath at 55 °C for 30 min. After incubation, 1 mL of PBS was added to each sample and centrifuged at 2000× *g* for 10 min. An amount of 300 mL of recovered supernatant was added to a 96-well microplate, and the absorbance was recorded at 560 nm. For a later evaluation of the morphological changes caused by the heat-induced damage to the RBC, the pellet (globular package) was stored at 4 °C. DS was used as standard at the same concentrations of the sample. Later, 300 μL of 10% RBC was used as Control (+), 150 μL of diclofenac sodium + 150 μL of 10% RBC was used as standard. The percentage of inhibition of hemolysis was calculated similarly to that in Equation (5). 

Hypotonicity-induced hemolysis was carried out according to the methodology of Agarwal et al. [22] with some modifications. Then 10% RBC suspension was prepared with PBS at pH 7.4. An amount of 50 μL of RBC suspension was mixed with 100 μL of PBS (phosphate buffer saline at 7.2 pH), 100 μL of microalgae extracts (300 μg·mL^−1^), and 200 μL of hyposaline solution. The mixture was incubated in a water bath at 37 °C for 30 min. After incubation, 850 μL of PBS was added to each sample, and the mixture was centrifuged at 2000× *g* for 10 min. Later 300 μL of recovered supernatant was added to a 96-well microplate. The absorbance was recorded at 560 nm. For a later evaluation of the morphological changes caused by the hypotonicity-induced damage to the RBC, the pellet (globular package) was stored at 4 °C. DS was used as standard at the same concentrations of the sample. Three controls were prepared as follows: (1) using 50 μL of 10% RBC + 300 μL of PBS, (2) using 50 μL of 10% RBC + 200 μL of PBS + 200 μL of hyposaline solution, and (3) using 50 μL of 10% RBC + 150 μL of diclofenac sodium + 200 μL of hyposaline solution. The percentage of the inhibition of hemolysis was calculated similarly to that in Equation (5). 

### 2.9. Antiproliferative Activity

For the proliferation assay, A549, HeLa, PC-3, LS-180, and ARPE-19 cells (all cell lines are human) were employed. The cell viability was assessed by 3-(4,5-dimethyl-2-thiazolyl)-2,5-diphenyltetrazolium bromide (MTT) reduction assay. All cell lines were cultivated in a 96-well sterile microplate with DMEM medium at a density of 10,000 cells/well. Each extract (acetonic, methanolic, and ethanolic) of *N. incerta* was dissolved in DMSO to obtain 25 to 200 μg·mL^−1^. DMSO was used as a control as established by the US National Cancer Institute (US-NCI), followed by 48 h of incubation. Ten microliters of MTT (5 mg·mL^−1^) were added to each well and incubated at 37 °C per 4 h. The living cells that will reduce MTT to formazan with mitochondrial enzymes were determined using an ELISA microplate reader Bio-Rad (Sigma-Aldrich Co., San Luis, MI, USA), at 570 and 630 nm. All of the experiments were carried out in parallel with a DMSO (0.06–0.50%) control by triplicate. The results were reported as IC_50_ (Concentration means inhibitory) values [30,31]. Then, morphological changes in the cancer and non-cancer cell lines were monitored for 48 h in an inverted microscopy (Olympus ckx41 inverted phase contrast tissue culture microscope; Thermo Scientific, Waltham, MA, USA).

### 2.10. Statistical Analysis

All data were analyzed using the statistical program JMP software v15 and R-Studio software v1.1.463 for Mac and expressed as mean (IC_50_) ± SD (standard deviation). Data were subjected to linear and non-linear regression and to a one-way analysis of variance (ANOVA) by comparing means with the Tukey test (*p* < 0.05). For association, an analysis was performed using the chi-square test to determine differences in proportion and the Mann-Whitney U test to compare independent groups in terms of metric variables. Odds ratios (ORs) were obtained by unconditional logistic regression with *p* < 0.05 confidence intervals. One and two-way ANOVA was performed to observe the interaction between the different association factors. The study was carried out under controlled conditions with a minimum of three repetitions for each analysis.

## 3. Results and Discussion

### 3.1. Acute Toxicity Bioassay on Artemia Salina

Table 2 shows the average of live and dead *Artemia salina* nauplii after exposure to *N. incerta* extracts and the extracts´ toxicity degree. It was observed that the ethanolic extract presented the lowest toxicity degree (46.67% ± 1.53), and it was considered relatively innocuous (LC_50_ > 1500 μg·mL^−1^). Similarly, the methanolic (LC_50_ = 1250 μg·mL^−1^) and acetonic (LC_50_ = 1500 μg·mL^−1^) extracts were classified as virtually non-toxic. These extracts could thus be used in the food and health industries. In this context, several studies were carried out using *A. salina* as a test model to classify the toxicity of extracts, some of them from microalgae, using different solvents [32,33]. 

The acute toxicity bioassay on *Artemia salina* nauplii revealed that the increase in the concentration of the acetone extract is not accompanied by a decrease in live cells at a concentration of 250 µg/mL (10 ± 0.00 live nauplii). This result is important for the study, since the acetone extract presents the highest hemolysis inhibition percentage for the different blood groups at concentrations of 333 µg/mL (point Section 3.3.1). However, this concentration was not tested in the acute toxicity bioassay. Probably, the concentration of 333 µg/mL would not significantly affect *A. salina*, since at 750 µg/mL a decrease of 1.3 ± 1.53 dead nauplii was observed. This indicates that the acetone extract would be harmless to an organism at these concentrations. This could be due to the protective effect provided by these extracts, which not only do not present toxicity, but can even protect organisms or cells such as erythrocytes from oxidative stress, agents that do promote cell death (see Section 3.3.1).

### 3.2. Blood Biocompatibility Assay of N. incerta Extracts on Different ABO and Rh Blood Groups

With the development of natural products based on medicine that provides health benefits, the question of their possible toxicity has gained attention. Through digestion, these compounds are released into the bloodstream; therefore, cytotoxic assays are required in blood, specifically on erythrocyte surface antigens (ABO and Rh blood system). This blood biocompatibility test evaluated the potential hemolysis percentage when the RBC are exposed to extracts from *N. incerta*. The occurrence of erythrocyte hemolysis could be an indicator of cell oxidative damage that leads to cell death [34]. According to the applied test, the methanolic extract did not show significant differences (*p* > 0.05) between the different ABO blood groups, presenting very low values of hemolysis (1.56% to 3.25) at concentration of 150 μg·mL^−1^ (Table 3). The blood groups most susceptible to hemolysis were B (14.73% ± 1.79 of hemolysis) and O (14.22% ± 1.12 of hemolysis) in the presence of the acetonic and ethanolic extract, respectively. 

The acetonic and methanolic extracts presented greater blood biocompatibility with the O RhD-ve group, with low hemolysis values (0.67 to 1.02, respectively) indicating low toxicity (Table 4). Meanwhile, blood group O RhD+ve is significantly more susceptible to hemolysis from the toxicity induced by all of the tested extracts (1 to 12.6% hemolysis) than the RhD-ve group. The degree of toxicity due to exposure to the extracts was ranked as follows: ethanolic extract > methanolic > acetonic. This result does not indicate that the degree of toxicity of the extracts should affect further studies of their use as a food additive or nutraceutical. However, the mechanism of pigment-induced hemolysis has not been thoroughly studied. More in-depth studies would have to be performed using flow cytometry and FACS analysis using Annexin V-FITC [35] to analyze the activity of eryptosis (apoptosis of erythrocytes) during the cytotoxic action of the compounds. Additionally, genetic studies must demonstrate that the genes involved in the synthesis of ABO and Rh antigens are associated. A multiparametric analysis technique would make it possible to identify the cells in eryptosis through the action of the apoptotic genes. Eryptosis occurs when the erythrocyte is too damaged to be viable, so the cell decides to enter its programmed death, which indicates that the compounds´ toxic damage is irreversible [36,37]. The involvement of ABO and Rh receptors with the susceptibility to hemolysis in the RBC membranes and their interaction with pigments has not yet been established. However, when Chisté et al. [34] tested β-cryptoxanthin, they found a significant cytotoxic effect at a concentration of 0.6 μM, but a concentration of 0.1 μM did not show some hemolytic effect. It should be noted the effect of a pure compound can be toxic at high concentrations compared to a crude extract. Thus, this cytotoxic effect is not expected to occur under normal conditions in the blood system because the presence of this carotenoid is very low in the blood (<0.03 μM) of healthy humans [38]. Eryptosis manifests itself under cytotoxicity and oxidative stress and exposure to xenobiotics [36,37]. This action is observed in the hyperosmolarity of some plant and algae phytochemical compounds (e.g., coumarins, phenols, flavonoids, tannins, or saponins). Results of the blood biocompatibility assay by percent hemolysis test in the present study could be considered for further in vivo studios for biomedical applications.

### 3.3. Association of Different ABO and Rh Blood Groups with the Erythroprotective Effect against Oxidative Stress

#### 3.3.1. Antihemolytic Activity

All of the extracts of *N. incerta* showed a high capacity to inhibit the AAPH-induced free radicals, avoiding erythrocyte membrane degradation (Table 5). The results indicated that the acetonic extract had the highest values (83–98%) of antihemolytic activity, showing significant differences between the ABO blood groups (*p* < 0.05). The effect on erythrocytes was enhanced in the presence of antigen A (~98%). The methanolic extract was the only one that did not show significant differences between the ABO blood groups (68–77%) (*p* > 0.05). The ethanolic extract exhibited a differential antihemolytic effect on human erythrocytes in the ABO group system (62–83%). The lysis of erythrocytes was found to decrease in the presence of antigen A (~62%). 

The result suggests that A group RBC are susceptible to lipid peroxidation, decreasing the erythroprotective effect of the ethanolic extract. Due to this, a percentage of erythrocytes are compromised and collapse due to the oxidation of hemoglobin and polyunsaturated fatty acids in the cell membrane. As for the Rhesus factor (Rh), the antihemolytic effect was higher in RhD+ve (86–94%) than RhD-ve (68–80%) in all samples tested in this study (Table 6). The hemolysis inhibition effects of the methanolic and ethanolic extracts on RhD+ve erythrocytes were significantly different (*p* < 0.001), where the most significant effect was registered in the methanolic extract (~94%). Although the expected results were favorable, the blood groups could have a link with the antiradical effects of bioactive compounds depending on the surface antigen. These results suggest that the selected solvent may potentially act as an important erythroprotective agent by preventing ROO•-induced hemolysis in human erythrocytes without compromising the structure of the cell membrane at risk. 

#### 3.3.2. Association Study by Univariate Logistic Regression

Univariate logistic regression was applied to show that the antihemolytic effect is surface-antigen-dependent. The strength of association between dependent (the pigments´ antihemolytic activity) and independent (the blood groups) variables was assessed using univariate logistic regression to show that the antihemolytic effect is surface-antigen-dependent (Table 7). The study demonstrated the association of the antihemolytic activity of the acetone and ethanolic extracts with the A blood group (OR = 1.19; 95% CI = 0.17–2.22 and OR = 2.60; 95% CI = 0.68–1.77, respectively) versus the B and O blood groups. In the meantime, an association was found between the antihemolytic activity in the three extracts (acetone, methanolic, and ethanolic) and the presence of the D antigen (OR = 2.61; 95% CI = 1.38–3.61, and OR = 2.17; 95% CI = 1.51–2.83, and OR = 1.47; 95% CI = 0.61–2.33, respectively). Meanwhile, the absence of antigen D was associated with antihemolytic activity only in the ethanolic extract (OR = 1.28; 95% CI = 1.25–1.6). These results demonstrate that the antihemolytic activity is increased in the presence of blood groups A, as they were significantly associated with the extracts. Our results point to the need for additional basic research on the role of antioxidant phytochemicals in the presence of erythrocyte surface antigens.

#### 3.3.3. Micrographs of Human Erythrocytes after Erythroprotective Effect

The erythroprotective effect from the *N. incerta* extracts was observed by optical microscopy through morphological changes in different ABO (Figure 1) and Rh (Figure 2) blood groups. Figure 1A–C and Figure 2A–C (erythrocytes without AAPH) show the typical morphology of a healthy human erythrocyte (ABO and Rh blood group) with an approximate size of 6–7.5 nm, which consists of a typical pale biconcave disk-shaped area, since a depression forms in the central area [39]. To describe the lethality of free radicals on ABO and Rh (O+ and O −) blood group membranes, Figure 1A–C and Figure 2A–C (erythrocytes with AAPH) are shown. Undoubtedly, all erythrocyte cell models used as hemolysis controls show free-radical-induced disruption of the membrane. Lipid peroxidation induced by peroxyl radicals generated by the thermic decomposition of AAPH promotes the loss of the integrity of the RBC membrane, exhibiting a typical hemolytic activity [40]. This hemolysis is characterized by perforations in the plasma membrane, an extraordinarily deformed cellular structure, and a color change that goes from deep red to deep purple. In the meantime, Figure 1A shows the erythroprotective effect of the acetonic extract on ABO blood cells. No significant morphological changes are observed in the ABO groups in the presence of the acetone extract. The erythroprotective effect is evidenced by the preservation of the integrity of the erythrocyte membrane. The pigments in the extract inhibited the radicals generated by the AAPH, avoiding their interaction with the membrane and their entry into the cytosol. Due to the above, the RBC shows the typical morphology of a healthy cell. On the other hand, the methanolic extract´s erythroprotective effect on ABO blood cells differed in each blood group (Figure 1B). The erythroprotective effect was insufficient since slight damage, almost non-noticeable, could be seen in the A+ and B+ red cell membranes. The membrane structure shows slight damage near the biconcave zone. The free radicals interact with the membrane lipids, causing a slight peroxidation. The presence of group O+ significantly reduces the erythroprotective potency of the methanolic extract. This decrease in protective activity allowed damage to the cell membrane. It is likely that the ROS-induced damage did not compromise the membrane stability enough to cause eryptosis. Figure 1C shows a significant reduction of the erythroprotective effect from the ethanolic extract in the A+ group (~62% of hemolysis) compared to the A+ and O+ groups (~80–83% of hemolysis). In this case, the antihemolytic effect decreased in the ethanolic extract due to the A+ antigen. The resulting oxidative stress damage to the membrane causes granulation, vacuoles, and perforations that release its content to the outside, which at a specific moment could cause eryptosis. Figure 2 shows the morphological changes caused by AAPH-induced hemolysis on the Rh of RBC (O+ and O− blood group). Oxidative damage was evident in O− erythrocytes compared to O+ in all extracts (Figure 2A–C). Therefore, the absence of an RhD antigen decreased the potency of all extracts. The membrane cells are susceptible to peroxidation due to the high concentration of highly polyunsaturated fatty acids [41]. However, to date, these studies do not specify the type of blood group used for the test but emphasize that only a single blood group is used. The possible associations of the different ABO and Rh blood groups with the erythroprotective and antihemolytic effects on human erythrocytes has not been reported. Therefore, our study is the first paper that analyzes a probable association of the different blood groups with antihemolytic activity. A previous study [20] demonstrated that the pigments produced by *Navicula incerta* have a potent antihemolytic activity (81.8–96.7% of hemolysis inhibition), preventing oxidative damage to the lipidic membrane of human erythrocytes.

The pigments’ antiradical activity occurs before they contact the plasma membrane. The interaction of the pigments with membrane surface glycoproteins (e.g., ABO antigens and immunoglobulins such as IgG and IgM) could form a complex to free-radical scavenging reducing the membrane oxidation [42]. The main pigments found in diatoms are fucoxanthin, astaxanthin, neoxanthin, zeaxanthin, and canthaxanthin, capable of inhibiting the hydroxyl and peroxyl radicals generated by the AAPH [43,44,45]. Studies by Chisté et al. [34] observed that the basic structure of these compounds has a crucial role in their antihemolytic effect. The carotenoids consist of a system of conjugated double bonds (CDB), which generates a resonance system of π electrons that move through the hydrocarbon chain of isoprenoid polymers. In contrast, the hydroxyl group can transfer hydrogen atoms in xanthophylls [46,47,48].

Research suggests the probability that these blood groups are also associated with the systemic inflammatory response that is involved in the development of different types of cancer [3,4,49,50,51], in particular among people with genotype “A” [52]. They have also, however, been associated with the risk of bacteria and virus transmission infections, e.g., infection with HIV, hepatitis B virus [53], and severe acute respiratory syndrome coronavirus 2 (SARS-CoV-2) [9]. Today, a new deadly virus named coronavirus-2019 (COVID-19) has reached the whole world and was declared a global pandemic by the World Health Organization (WHO) [54]. Zhao et al. [55] and Zietz et al. [56] found a relationship between the “A” blood type and susceptibility to COVID-19. Blood group “A” is associated with a 50% increased risk of needing respiratory support in the case of coronavirus infection. Conversely, having blood group “O” confers a protective effect against the development of respiratory failure (35% less risk) [9,57,58]. For this reason, our study has focused on determining the antihemolytic and erythroprotective effects of the pigments on RBC, aiming to prevent peroxidation by radical inhibition. The results indicate that the pigments in the acetonic extract could be potential candidates for conferring a protective effect on the RBCs of group A, preventing the risk of these diseases associated with antigen A. In the future, functional foods could be developed aimed at these health sectors. Compounds derived from the diatom *N. incerta* could be candidates for preventing the development of cancer and infection-borne diseases in relation to particular blood groups.

### 3.4. In Vitro Anti-Inflammatory Activity Screening

#### 3.4.1. Inhibition of PPE Enzyme Assay

The anti-inflammatory activity of the *N. incerta* extracts was evaluated by inhibiting the elastase activity (Figure 3), and the percentage of inhibition and half-maximal inhibitory concentration (IC_50_) were measured (Table 8). All extracts and DS showed a decrease in their absorbance during the reaction time, relative to the control (enzyme + substrate). The decrease in the absorbance is directly proportional to the p-nitroaniline release. The acetonic extract showed a high inhibition of PPE (96.64% ± 4.82) compared to methanolic extract (72.57% ± 2.92), ethanolic extract (62.39 ±1.51%), and DS (61.18% ± 2.23). In the same way, the acetonic extract showed a lower IC_50_ value (10.82 ± 3.24 µg·mL^−1^) than the methanolic (34.69 ± 4.14 µg·mL^−1^) and ethanolic (35.39 ± 4.12 µg·mL^−1^) extracts and DS (35.75 ± 2.56 µg·mL^−1^). Hence, all of the extracts and DS presented anti-inflammatory activity by elastase inhibition. DS is an anti-inflammatory commonly used as an analgesic and antipyretic in treating acute rheumatic diseases and rheumatoid arthritis. However, recently the diclofenac solution for injection was banned due to its severe nephrotoxicity due to the use of Transcutol-P as a solubilizing agent. Therefore, new compounds with anti-inflammatory activity are being sought. Thus, it is important to find compounds that exhibit anti-inflammatory activity and do not compromise people’s health. The extracts of *N. incerta* could contain anti-inflammatory carotenoids such as astaxanthin and β-carotene, which might fulfil this function. These compounds are powerful anti-inflammatories beneficial to health, preventing or treating pathologies such as chronic inflammation, as in as arthritis, for example. However, no information on the mechanism of the inhibition of pigments for elastase-like enzymes has been found in the literature. The mechanisms of PPE inhibition have not been studied using extracts derived from the microalga *N. incerta* with a high carotenoid content. Brás et al. [59] observed that polyphenols (antioxidants) bind to amino acids such as arginine glycine, alanine, serine, valine, and leucine, which make up the elastase, by covalent bonds and have a reversible competitive inhibition on the PPE enzyme. Hence, the pigments (also antioxidants) probably are attached by covalent bonds to elastase, enhancing the elastase´s affinity to N-s(ala)pN and preventing the release of p-nitroaniline. These mechanisms are necessary to exert anti-inflammatory activity by enzyme inhibition. For this reason, more studies are needed to determine the mechanism of inhibition exerted by the pigments on the PPE.

#### 3.4.2. Inhibition of Albumin Denaturation Assay (Antarthritic Activity)

It has been documented that protein denaturation is well-implicated in developing an inflammatory process in cells. Several chronic diseases are related to protein denaturation, with arthritis being one of them. Some research has looked for alternative ways of preventing this type of inflammation [12]. Table 4 shows the inhibitory effect on bovine serum albumin (BSA) denaturation by the *N. incerta* extracts. The extracts were moderately (below 50% inhibition) effective in inhibiting the heat-denaturation of albumin at the concentrations tested (200 µg·mL^−1^). No significant differences (*p* > 0.05) were observed among the extracts analyzed (39–43%, approximately). Unfortunately, the antiarthritic effect of the extracts did not exceed DS (94.67% ± 2.61 at a concentration of 200 µg·mL^−1^), being more effective in inhibiting the denaturation of the BSA. Rheumatoid arthritis is a multisystem inflammatory autoimmune disease that affects surrounding joints and promoted cartilage destruction. Eventually, it can lead to stiffness, deformity, and disability [55].

The immune system intervenes during a normal anti-inflammatory response by releasing pro-inflammatory proteins and molecules to reduce inflammation. However, in inflammatory arthritis, the immune system attacks the joints, causing swelling, stiffness, and pain. Therefore, powerful anti-inflammatory drugs such as DS are required to counteract these symptoms [60]. This drug is a potent anti-inflammatory that inhibits the denaturation of the protein and is commonly used to treat diseases associated with arthritis [12]. Although DS was more effective in inhibiting BSA’s denaturation, it is known to cause adverse effects on health [61]. Previous research has demonstrated that pigments produced by Chlorella vulgaris possess significant anti-rheumatoid arthritis/anti-inflammatory potential, as they prevent the denaturation of BSA (68% at 500 µg·mL^−1^) similarly to synthetic drugs such as aspirin (71% at 500 µg·mL^−1^) [62,63]. In our study, all extracts were tested at a concentration of 500 µg·mL^−1^, and that could make a difference: the greater the concentration of the extract, the more the protein denaturation increases. BSA is used as a protein model since it is found at a concentration of 60% in human serum. When this protein is exposed to prolonged heat, it expresses type III hypersensitivity associated with the antigens related to chronic inflammations, such as glomerulonephritis, rheumatoid arthritis, systemic lupus erythematosus, and serum sickness. On the other hand, some studies have associated the pathogenesis of rheumatoid arthritis disorder with the free radicals-induced oxidative stress [22,62]. Therefore, the antihemolytic activity (that is considered an antioxidant property) of the extracts from *N. incerta* could contribute to the inhibition of inflammation-associated arthritis.

#### 3.4.3. Membrane-Stabilizing Capacity Assay

The use of the human red blood cell (HRBC) membrane to evaluate the stability of the lysosomal membrane bioactive compounds has become increasingly common. The erythrocyte membrane is similar to the lysosomal membrane, which comes from activated neutrophils, and its stabilization can be extrapolated to the stabilization of the lysosomal membrane [64]. In terms of chronic inflammation, the membrane-stabilizing capacity of the bioactive compound is essential. The lysosomal membrane´s denaturation releases hydrolytic and proteolytic enzymes, which are responsible for degrading fibrosis tissue and also cause heterophagy and autophagy [64].

All the extracts tested showed a high percentage of heat-induced hemolysis inhibition (100%) (Table 8). The membrane-stabilizing capacity of the extracts was higher (*p* < 0.05) than the inhibition percentage of the DS (92.32% ± 8.06), which was used as a standard. The compounds found in all of the extracts of *N. incerta* play an important role as protective agents in stabilizing the biological membranes, including stabilizing the lysosomal membrane against thermal lysis. The stabilization of this organelle is relevant to limiting the inflammatory response and inhibiting the hydrolytic enzymes that cause inflammation in cell tissue. This action limits the inflammatory response by inhibiting markers and constituents, such as proteases [64]. However, the anti-inflammatory mechanism carried out by this method for pigments such as chlorophylls and carotenoids derived from microalgae has not been reported in the literature.

A hypotonicity-induced hemolysis assay was carried out to evaluate the anti-inflammatory properties of the *N. incerta* extracts (Table 8). The ethanolic extract (91.74% ± 0.51) and DS drug (93.80% ± 0.88) effectively inhibited the hypotonicity-induced hemolysis of RBC at the extract concentrations tested (300 µg·mL^−1^). The results show that the ethanolic extract could be used as an anti-inflammatory agent, with the same ability to reduce inflammation by releasing lysosomal compounds. Hypotonicity-induced hemolysis is another way to evaluate the membrane stabilization capacity [64]. The fluid accumulation inside the cell makes the membrane susceptible to reactive molecules such as superoxide (O_2_•), hydroxyl (HO•), and peroxyl (ROO•), radicals that produce lipid peroxidation. The mechanism of pigments’ action could be associated with their ability to limit the free-radical scavenging presented by the extracts from *N. incerta*. This action could play a very important role in its membrane-stabilizing capacity. Studies carried out by Prabakaran et al. (2018) [62] with methanolic extracts of *Chlorella vulgaris* have shown protective activity on the RBC membrane, inhibiting hemolysis (70%) at 500 µg·mL^−1^, lower activity than in our study. Additionally, studies based on microalgae show that they can intervene in the release of neutrophil lysosomal content at the time of the inflammatory response [65,66]. Some benthic diatoms, such as *N. incerta*, produce polar and non-polar carotenoids that could be responsible for the inhibition of membrane denaturalization. Lutein, zeaxanthin, astaxanthin, zeaxanthin, and β-carotene exhibited significant anti-inflammatory activity, preserving the integrity of the membrane [17,18]. These compounds probably exhibit the protective effect with a similar mechanism to the one shown in antihemolytic activity.

#### 3.4.4. RBC Morphological Changes Induced by Heat and Hypotonicity Damage by Optical Microscopy

As shown in Figure 4, the extract induced an erythroprotective effect on the membrane against thermal and hypotonic lysis. Figure 4A (control without heat application) and F (control without hypotonicity solution application) show a typical healthy erythrocyte morphology with a normal size (6–7.5 nm), regular biconcave shape, and depressed central zone, presenting no abnormalities in the membrane. In contrast, the denaturation of the erythrocyte membranes and their hemolysis is evident in Figure 4E (thermal lysis) and J (hypotonic lysis), which were used as a hemolysis control. In these events, where the protective effect did not occur, lipids begin to be oxidized by free radicals in the lysosomal membrane, causing an inflammatory response. The release of proteolytic enzymes is responsible for fibrosis tissue degradation and promotes the inflammation-associated arthritis disorder [64,65]. In this case, the lysosome cell lysis releases inflammation markers such as the C reactive protein, interleukins, and pro-inflammatory proteins. In higher concentrations, these molecules cause chronic inflammation, leading to a carcinogenesis process [67,68,69].

Figure 4B–D show the protective effect of all of the extracts on erythrocytes subject to heat-induced hemolysis. Despite being in a temperature condition of 55 °C, the erythrocyte showed a typical morphology with normal size (6–7.5 nm) and do not present any abnormalities in the plasma membrane. This erythroprotective effect could be due to a biofilm formation, highlighting a slight green color that envelops the whole cell. The pigments could be coating the cell membrane, inhibiting hemolysis and lipid oxidation and fulfilling their membrane-stabilizing capacity. The ethanolic extract showed the highest protective effect against hypotonicity-induced hemolysis (Figure 4I), exhibiting healthy erythrocytes with a cell regular morphology and without degradation of the membrane. In these samples, the presence of the pigment is stronger, and the biofilm adheres to the membrane due to a greater reinforcement that maintains the cellular osmoregulation. Meanwhile, the protective effect in the acetonic and methanol extracts (Figure 4G and Figure 4H, respectively) was less powerful. The pigments accumulated in the extracts of *N. incerta* could act as antioxidant agents [16], reducing free radicals and preventing the activation of lysosomal neutrophils. In addition, the inhibition of the release of lysosomal content at the site of inflammation can also explain the anti-inflammatory properties of the extracts [12]. These results suggest that *N. incerta* pigments are capable of inhibiting the release of arthritis-inducing components and could have potential anti-inflammatory applications.

### 3.5. Antiproliferative Activity

The results of the antiproliferative activity of the *N. incerta* extracts assayed against four different cancer cell lines (A549, HeLa, PC-3, and LS-180) and one non-cancer cell line (ARPE-19) are shown in Table 9. The test revealed that the pigmented extracts from *N. incerta* have the highest cytotoxic activity in two different cancer cell lines (HeLa and PC-3) without affecting healthy cells (ARPE-19). In the MTT reduction assay, the acetonic, methanolic, and ethanolic extracts were tested over 25–200 μg·mL^−1^. The ethanol extract on the HeLa and PC-3 cells lines presented an IC_50_ value of 59.28 ± 2.58 and 96.05 ± 3.48 μg·mL^−1^, respectively. The National Cancer Institute of the United States (NCI-US) considers IC_50_ values lower than 200 μg·mL^−1^ in the raw extracts to be good antiproliferative activity [70]. The phytochemical components most likely to exert the cytotoxic effect may be the carotenoids, since they can intervene in the death mechanisms of cells. Carotenoids like neoxanthin, fucoxanthin, and fucoxanthinol revealed an antiproliferative activity involved in the cell cycle mechanism, apoptosis, and metastasis [19]. They have a high effect on the growth of PC-3, starting the mechanism of programmed cell death (apoptosis) by activating the caspase-3 and increasing mortality above 30% [19,67,71,72]. β-carotene and astaxanthin are important carotenoids for the inhibition of colon cancer proliferation. These carotenoids are probably found in low concentrations in the extracts of *N. incerta* since the proliferation of colon carcinoma was not inhibited [19,72,73]. Recently, it has been reported that astaxanthin inhibits the enzyme 5-α-reductase involved in the atypical growth of the mouse prostate in in vivo assay [73]. This pigment has been isolated in different microalgae species such as Haematoccus pluvialis; however, in *N. incerta* it has not been reported. On the other hand, carotenoids such as β-cryptoxanthin can regulate the differentiation of the expression of p73 variants, which are involved in cell cycle regulation, cell death, and senescence. Studies have shown that this regulation is linked to the inhibition of the proliferation of prostate cancer, which can induce apoptosis negatively regulated by the ΔNP73 [67,74].

The cytotoxic effects of the different extracts obtained from *N. incerta* on the PC-3, HeLa, and ARPE-19 cell lines at three different exposure times, 0 (control), 24, and 48 h, were observed using the microscopy technique (Figure 5). Morphological changes in HeLa and PC-3 cell lines were evident after exposure to the cytotoxic agent (extracts) at 200 μg·mL^−1^, and, after 24 h, significant effects on cell viability could be observed. HeLa, PC-3, and ARPE-19 control cell lines were presented within the normal morphological limits (Figure 5A–C). A variation in the size and shape typical of cell division and uncontrolled maturation is observed. Anisocytosis (abnormal nucleus size), round, oval, and bean-shaped cells occur for both cell cancer lines. Chromatin is shown to be increased, but its distribution and quantity vary from one nucleus to another due to uncontrolled pre-proliferation. HeLa and PC-3, in particular, display abnormal growth patterns, because they can evade programmed cell death (apoptosis) and senescence. Because cells acquire the ability to divide multiple times, these characteristics are usually present in cancer cells such as HeLa and PC-3 [74]. Typical healthy retinal cell morphology (of an ARPE-19 non-cancer cell) was observed at 0 h (control) of exposition at the microalgal extracts. The typical characteristics of healthy retinal cells, defined as cell borders, good pigmentation, and a “cobblestone” appearance, are present. A single epithelial monolayer is observed, similar to Dunn et al. [75] and Hazim et al. [76].

The cytotoxic effect of the ethanolic extract on PC-3 and HeLa at 24 and 48 h (Figure 5D,E,G,H) decreased the cellular viability and induced morphological changes, pyknosis, and loss of adhesion, causing sporadic distribution at a concentration of 200 μg·mL^−1^. In addition to condensed chromatin and the formation of apoptotic bodies, karyorrhexis around the plasmatic membrane was also observed, consisting of the separation of the apoptotic bodies. Another characteristic observed was onychosis, cells that have large nucleus lipid peroxidation and proteolysis, similar to Fink et al. [77]. The results suggest that the ethanolic extract could induce apoptosis in the PC-3 and HeLa cells. However, apoptosis induction results must be confirmed in subsequent studies by the TUNEL assay, which detects the formation of DNA fragmentation [77]. On the other hand, cells with autophagy characteristics showed organelles and cytoplasm by lysosome-borne neighbors for their degradation. The extract effect was not reflected in the same way for ARPE-19 as for HeLa and PC-3 (Figure 5F,I) after 48 h in contact with the extracts. Generally, throughout this period, the ARPE-19 cell culture is darker. The culture shows a stable cell organization with the typical structural morphology of healthy cells [76]. Morphological changes in cell lines are caused by the antiproliferative effect of pigment compounds (carotenes and xanthophylls) from the *N. incerta* extracts. Probably, fucoxanthin and fucoxanthinol are involved in the process of cell cycle arrest; this could suggest that the cell death mechanism consists basically of apoptosis [19,78,79].

The effects on cell viability are significant, potentially inhibiting cancer cell proliferation. This effect promotes increased apoptosis and cell cycle arrest. Studies have observed that fucoxanthin promotes DNA fragmentation, while carotenoids such as astaxanthin and β-carotene do not significantly affect DNA fragmentation [79]. The cleavage of caspases-3 and 9, together with poly-ADP-ribose polymer (PARP), is essential to astaxanthin´s cytotoxicity mechanism in cancer cell lines [78,80]. In some studies, fucoxanthin, neoxanthin, phytoene, and zeaxanthin were found to be involved in decreased PC-3 cell viability, where neoxanthin exhibited the highest rate of proliferation of the ingestion of this cell line. The extracts and their metabolites could be used as chemotherapeutic agents administered during the early stages of prostate and cervical cancer [78,80,81,82].

## 4. Conclusions

The study revealed that *N. incerta* extracts were considered virtually non-toxic. The data indicate that the extracts would be suitable to use, as they showed no toxicity in tests with *A. salina*. The extracts show high blood biocompatibility. Therefore, there is no apparent damage to the erythrocyte plasma membrane. The expression of ABO antigens on the surface of erythrocytes demonstrates an association with the erythroprotective effect against oxidative damage. These results suggest that the selected solvent and the compounds from *N. incerta* may potentially act as important erythroprotective agents by preventing ROO•-induced hemolysis in human erythrocytes on the A and RhD+ve antigens without compromising the structure of the cell membranes at risk. However, more prospective and laboratory studies are needed to define how different antigens influence the effects of bioactive compounds. The erythroprotective effect also has anti-inflammatory properties, shown by its stabilization of the erythrocyte membrane, since it is analogous to the lysosomal membrane. Stabilizing it is relevant to limiting the inflammatory response and inhibiting the release of the lysosomal constituents of activated neutrophils. The ethanolic extract of *N. incerta* inhibited the proliferation of cervical (HeLa) and prostate (PC-3) carcinoma. This approach to the combinatorial analysis of different biological activities can help design phytochemicals as new candidates for preventive drugs to treat the chronic degenerative diseases associated with a specific blood group. It could lead to the development of functional foods designed specifically for people with a specific blood type and yield better results when treating diseases of this kind. Thus, these molecules could be a potential alternative for biotechnological or pharmacological applications, as they exhibit anti-inflammatory, antiproliferative, antiradical, erythroprotective, and antihemolytic effects.

## Figures and Tables

**Figure 1 metabolites-12-01203-f001:**
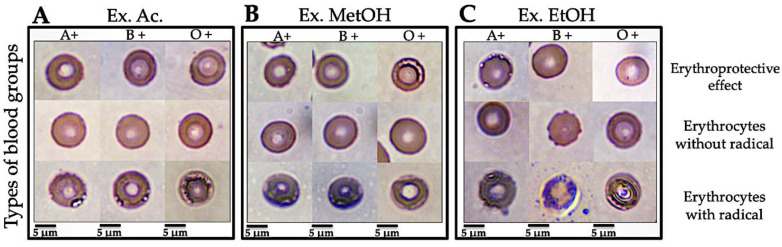
Micrographs of human erythrocytes (100×). Use of red blood cell membranes to evaluate the protective effect against oxidative damage from *N. incerta* extracts in different blood group types (ABO system). Morphological change seen in micrography is caused by free-radical-induced hemolysis (using AAPH molecule as a free radical generator). (**A**) Protective effect of the acetone extract (Ext. Ac.) of *N. incerta* on ABO blood groups. (**B**) Protective effect of the methanolic extract (Ext. MetOH.) of *N. incerta* on ABO blood groups. (**C**) Protective effect of the ethanolic extract (Ext. EtOH.) of *N. incerta* on ABO blood groups. Barr = 5 μm.

**Figure 2 metabolites-12-01203-f002:**
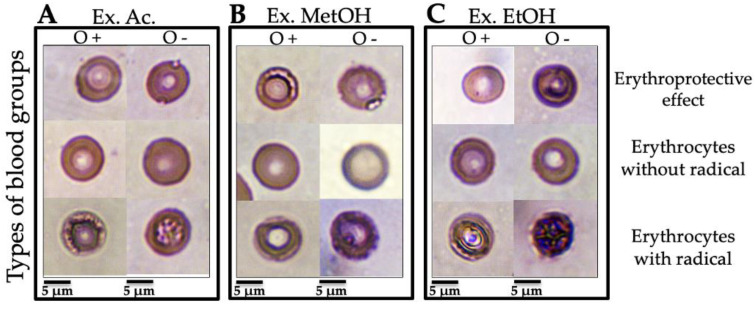
Micrographs of human erythrocytes (100×). Use of red blood cell membranes to evaluate the protective effect against oxidative damage from *N. incerta* extracts in different Rh (Rh+ and Rh−. Morphological change seen in micrography is caused by free-radical-Induced hemolysis (using AAPH molecule as a free radical generator). (**A**) Protective effect of the acetone extract (Ext. Ac.) of *N. incerta* on ABO blood groups. (**B**) Protective effect of the methanolic extract (Ext. MetOH.) of *N. incerta* on ABO blood groups. (**C**) Protective effect of the ethanolic extract (Ext. EtOH.) of *N. incerta* on ABO blood groups. Barr = 5 μm.

**Figure 3 metabolites-12-01203-f003:**
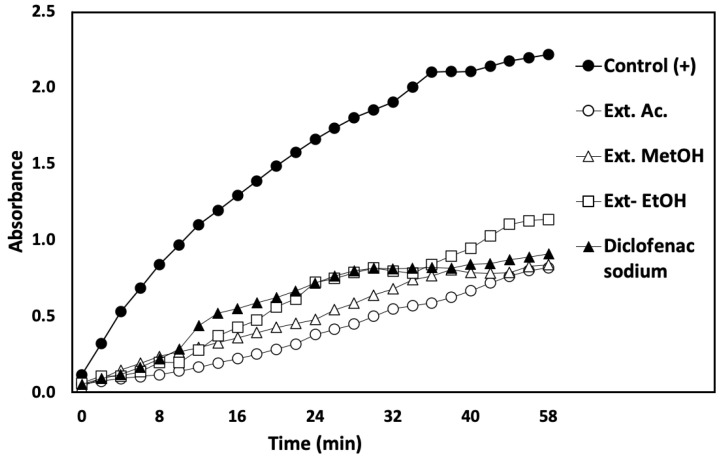
Anti-inflammatory activity evaluated by inhibition of elastase activity. Effect of the extracts on the inhibition of PPE. The absorbance increases when the p-nitroaniline is released. Control (+) Enzyme + substrate. Diclofenac sodium was used as anti-inflammatory control. The concentration of the extracts was 66.66 μg·mL^−1^.

**Figure 4 metabolites-12-01203-f004:**
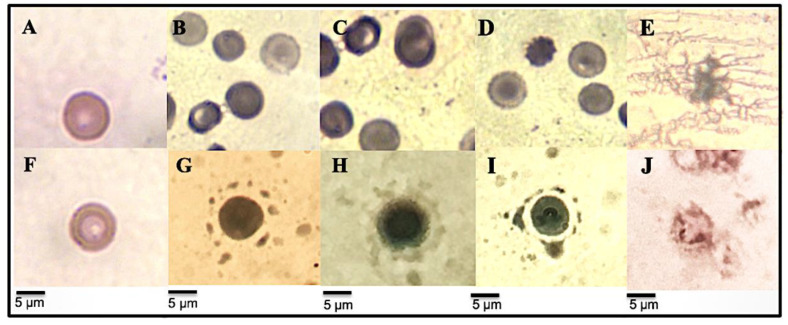
Morphological change micrography of human erythrocytes caused by different methods of induction of hemolysis to evaluate anti-inflammatory activity. Heat-Induced hemolysis: (**A**) Healthy erythrocytes; (**B**–**D**) erythrocytes + extracts (acetonic, methanolic, and ethanolic extract, respectively) + 55-degree hot water bath; (**E**) erythrocytes subjected to a temperature of 55-degrees in a hot water bath. Hypotonicity-Induced hemolysis: (**F**) Healthy erythrocytes; (**G**–**I**) erythrocytes + extracts (Acetonic, methanolic and ethanolic extract, respectively) + hypotonic solution; (**J**) erythrocytes + hypotonic solution. Micrograph to ×100. The bar represents a scale of 5 μm erythrocytes + hypotonic solution.

**Figure 5 metabolites-12-01203-f005:**
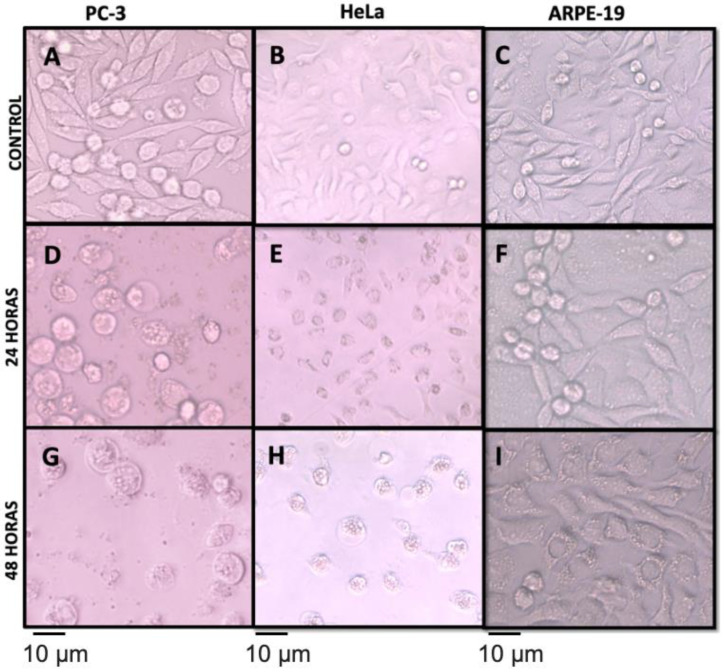
Effect of the different extracts obtained from *N. incerta* on the cell viability of A549, HeLa, and ARPE-19 cell lines during 48 h at a concentration of 200 μg·mL^−1^. Cells with nuclear/condensed chromatin fragmentation, apoptotic bodies, and “bleb” membranes are appreciated. Micrograph at ×50. Cell viability of PC-3 (**A**), HeLa (**B**), and ARPE-19 (**C**) at 0 h culture time (Control). Cell viability of PC-3 (**D**), HeLa (**E**), and ARPE-19 (**F**) at 24 h culture time. Cell viability of PC-3 (**G**), HeLa (**H**), and ARPE-19 (**I**) at 24 h culture time. The bar represents a scale of 5 μm.

**Table 1 metabolites-12-01203-t001:** Classification of extracts’ toxicity to *A. salina*.

* Classification	μg·mL^−1^
(E) Extremely toxic	1–10
(H) Highly toxic	10–100
(M) Moderately toxic	100–500
(S) Slightly toxic	500–1000
(V) Virtually non-toxic	1000–1500
(R) Relatively innocuous	>1500

* This classification is based on the mean lethal concentration (LC_50_) determination, according to Robles-García et al. (2016).

**Table 2 metabolites-12-01203-t002:** Toxicity bioassay on *Artemia salina* nauplii of *N. incerta* extracts at different concentrations.

Concentration (µg/mL)	Live Nauplii			Dead Nauplii			Toxicity Degree (%)		
	Ex. Acet.	Ex. MetOH	Ex. EtOH	Ex. Acet.	Ex. MetOH	Ex. EtOH	Ex. Acet.	Ex. MetOH	Ex. EtOH
5	9.7 ± 0.58	9.7 ± 0.58	9.3 ± 1.15	0.33 ± 0.58	0.33 ± 0.58	0 ± 0	3.33 ±0.58	3.33 ± 0.58	0 ± 0
50	9.7 ± 0.58	10 ± 0.00	10 ± 0.00	0.33 ± 0.58	0 ± 0	0.67 ± 1.15	3.33 ± 0.58	0 ± 0	6.67 ± 1.15
250	10 ± 0.00	9.7 ± 0.58	7 ± 2.00	0 ± 0	0.33 ± 0.58	1.67 ± 0.58	0 ± 0	3.33 ± 0.58	16.67 ± 0.58
750	8.7 ± 0.53	8.7 ± 0.58	8.3 ± 0.58	1.33 ± 1.53	0.33 ± 0.58	3.00 ± 2.00	13.33 ± 1.53	13.33± 0.58	30.00 ± 2.00
1250	6.7 ± 1.15	5 ± 3.00	6 ± 1.00	3.33 ± 1.15	5.00 ± 3.00	3.67 ± 0.58	33.33 ± 1.15	50.00 ± 3.00 *	36.67 ± 0.58
1500	5 ± 1.73	3.3 ± 2.52	6.3 ± 0.58	5.00 ± 1.73	6.67 ± 2.52	4.67 ± 1.53	50.00 ± 1.73 *	66.67 ± 2.52	46.67 ± 1.53 **

* (V) Virtually non-toxic (1000–1500 µg/mL); ** (R) Relatively innocuous (>1500 µg/mL); LC_50_ = mean lethal concentration; LC_50_ of Ex. Acet. = 1500 µg/mL; LC50 of Ex. EtOH > 1500 µg/mL.

**Table 3 metabolites-12-01203-t003:** Effect of the direct hemolysis from *N. incerta* extracts on different ABO blood groups.

Blood Group	Hemolysis (%)		
	Ex. Ac.	Ex. MetOH	Ex. EtOH
A	10.77 ^b^ ± 3.12	2.34 ^a^ ± 3.39	6.76 ^b^ ± 1.86
B	14.22 ^c^ ± 3.12	1.56 ^a^ ± 1.42	2.34 ^a^ ± 1.75
O	1.28 ^a^ ± 1.26	3.25 ^a^ ± 3.39	14.73 ^c^ ± 1.79

Values are mean ± standard deviation of at least three repetitions (*n* > 3). Means followed by different lowercase letter in the same column are different at a significance level of 0.05 (Tukey post hoc test). The concentration of the extracts was 150 µg·mL^−1^.

**Table 4 metabolites-12-01203-t004:** Effect of the direct hemolysis from *N. incerta* extracts on erythrocytes O RhD+ve and RhD-ve.

Blood Group	Hemolysis (%)		
	Ex. Ac.	Ex. MetOH	Ex. EtOH
O RhD+ve	1.35 ^a^ ± 1.23	4.13 ^b^ ± 1.08	12.58 ^b^ ± 3.48
O RhD-ve	0.67 ^a^ ± 0.15	1.02 ^a^ ± 1.27	5.91 ^c^ ± 2.87

Values are mean ± standard deviation of at least three repetitions (*n* > 3). Means followed by different lowercase letter in the same column are different at a significance level of 0.05. All data were analyzed using the *t*-Student test. The concentration of the extracts was 150 µg·mL^−1^.

**Table 5 metabolites-12-01203-t005:** Antihemolytic activity of *N. incerta* extracts against the oxidative hemolysis induced by AAPH on different ABO blood types.

Blood Group	Hemolysis Inhibition (%)		
	Ex. Ac.	Ex. MetOH	Ex. EtOH
A	88.33 ^b^ ± 2.98	68.67 ^b^ ± 5.36	83.52 ^a^ ± 2.06
B	98.61 ^a^ ± 3.03	76.33 ^a^ ± 3.51	62.18 ^b^ ± 7.61
O	83.78 ^c^ ± 2.78	77.73 ^a^ ± 4.03	81.21 ^a^ ± 3.09

Values are mean ± standard deviation of at least three repetitions (*n* > 3). Means followed by a different lowercase letter in the same column are different at a significance level of 0.05 (Tukey post hoc test). The concentration of the extracts was 333 µg·mL^−1^.

**Table 6 metabolites-12-01203-t006:** Antihemolytic activity of *N. incerta* extracts against the oxidative hemolysis induced by AAPH on erythrocytes O RhD+ve and RhD-ve.

Blood Group	Hemolysis Inhibition (%)		
	Ex. Ac.	Ex. MetOH	Ex. EtOH
O RhD+ve	88.16 ^a^ ± 0.69	69.09 ^a^ ± 2.44	86.08 ^a^ ± 3.23
O RhD-ve	68.21 ^b^ ± 4.93	68.18 ^a^ ± 3.21	80.51 ^b^ ± 2.51

Values are mean ± standard deviation of at least three repetitions (*n* > 3). Means followed by different lowercase letter in the same column are different at a significance level of 0.05. All data were analyzed using the t-Student test. The concentration of the extracts was 333 µg·mL^−1^.

**Table 7 metabolites-12-01203-t007:** Association of ABO blood group and Rhesus factor (Rh) with antihemolytic activity of extract from *N. incerta*.

Ex-Acet.		Ex-MetOH	Ex-EtOH
Blood Groups	OR (95% Cl)	OR (95% Cl)	OR (95% Cl)
O	0.29 (0.14–0.46)	0.45 (0.43–0.47)	0.91 (0.83–0.98)
A	1.19 * (0.17–2.22)	0.43 (0.42–0.45)	2.60 * (0.68–1.77)
B	0.51 (0.20–0.81)	0.52 (0.48–0.54)	0.73 (0.62–0.83)
RhD+ve	2.61 * (1.38–3.61)	2.17 * (1.51–2.83)	1.47 * (0.61–2.33)
RhD-ve	0.56 (0.17–1.0)	0.51 (0.47–0.53)	1.28 * (1.25–1.6)

* statistically significant at the level α = 0.05; OR, univariate logistic regression odds ratio; 95% CI, 95% confidence interval. All data were evaluated at 95% CI by logistic regression.

**Table 8 metabolites-12-01203-t008:** In vitro anti-inflammatory activity screening by inhibition of heat-induced hemolysis and inhibition of hypotonicity-induced hemolysis of extracts from the microalga *Navicula incerta*.

Samples	Inhibition of PPE (%)	IC_50_ to PPE (µg·mL^−1^)	Inhibition of BSA Denaturation (%)	Membrane Stabilization (%)	
				Inhibition of Heat-Induced Hemolysis	Inhibition of Hypotonicity-Induced Hemolysis
Ex. Acet.	96.64 ^a^ ± 4.82	10.82 ^a^ ± 3.24	43.92 ^b^ ± 3.23	100.12 ^a^ ± 1.23	87.91 ^b^ ± 1.35
Ex. MetOH	72.57 ^b^ ± 2.92	34.69 ^b^ ± 4.14	38.12 ^b^ ± 3.96	100.91 ^a^ ± 1.18	87.32 ^b^ ± 1.34
Ex. EtOH	62.39 ^c^ ± 1.51	35.39 ^b^ ± 2.56	39.77 ^b^ ± 1.64	100.74 ^a^ ± 1.68	91.74 ^a^ ± 0.51
DS	61.18 ^c^ ± 2.23	35.75 ^b^ ± 4.12	94.67 ^a^ ± 2.61	92.32 ^b^ ± 8.06	93.80 ^a^ ± 0.88

The data represent the mean ± standard deviation of at least three analyses. Different letters mean significant differences (*p* ≤ 0.05). One-way analysis of variance (ANOVA) with multiple comparisons of Tukey test. PPE = Pancreatic Porcine Elastase; herein, all samples’ concentrations were tested at 66.66 µg·mL^−1^. BSA = Bovine Serum Albumin. All samples’ concentrations were tested at 200 µg·mL^−1^ to BSA and 300 µg·mL^−1^ in membrane stabilization assays. DS = diclofenac sodium.

**Table 9 metabolites-12-01203-t009:** Antiproliferative activity (IC_50_ values) of extracts from *N. incerta* on human cell lines.

Extracts	IC_50_ μg·mL^−1^ ± SD				
	A549	HeLa	PC-3	LS-180	ARPE-19
Acetonic	>200 *	75.91 ^b^ ± 2.47	122.58 ^b^ ± 2.86	>200 *	>200 *
Methanolic	141.99 ^b^ ± 3.23	61.93 ^a^ ± 3.10	107.01 ^a^ ± 2.49	>200 *	>200 *
Ethanolic	123.73 ^a^ ± 2.59	59.28 ^a^ ± 2.58	96.05 ^a^ ± 3.48	>200 *	>200 *

Data are shown as the mean (IC_50_) ± SD (standard deviation) from at least three analyses. IC_50_ = Half-maximal inhibitory concentration. The different letter means significant differences (*p* ≤ 0.05). One-way analysis of variance (ANOVA) with multiple comparisons of Tukey test. The asterisk (*) represents the maximum concentration tested that did not reach IC_50_ values.

## Data Availability

The original contributions data presented in this research are included in the article; further inquiries can be directed to the corresponding authors.

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
