# Peer review of "Association of Different ABO and Rh Blood Groups with the Erythroprotective Effect of Extracts from Navicula incerta and Their Anti-Inflammatory and Antiproliferative Properties"

_metabolites, 2022, doi:10.3390/metabo12121203_

Round 1

Reviewer 1 Report

1. In Abstract clarify the sentence "Research suggests the probability that the blood groups are also associated with phytochemicals capable of mitigating oxidative stress related with systemic inflammatory response that is involved in the development of different types of cancer."

2. Line 72 rectify the error in the sentence "antiproliferative [19]. (Kumar 72 et al., 2013), and erythroprotective [20] agents." Line 75 "and neoplastic [6,7,8,9]. (Xia et al., 2018) and hemolytic diseases [20]." Line 535 "N. incerta with a high carotenoid content (Rajapakse et al., 2008). Brás et al. [59]"

3. Line 135-145. What are the composition of pigment extract? Have you identify the active ingredients of the extract that showed promising biological activity? I think more experimental work is need.

4. What is the chemical nature/composition/structure of the three extracts?

Author Response

Thanks to the reviewers for their comments.

  1. In Abstract clarify the sentence "Research suggests the probability that the blood groups are also associated with phytochemicals capable of mitigating oxidative stress related with systemic inflammatory response that is involved in the development of different types of cancer."

Response 1: The comment was considered. The sentence was clarified. See line 24-27.

  1. Line 72 rectify the error in the sentence "antiproliferative [19]. (Kumar 72 et al., 2013), and erythroprotective [20] agents." Line 75 "and neoplastic [6,7,8,9]. (Xia et al., 2018) and hemolytic diseases [20]." Line 535 "N. incerta with a high carotenoid content (Rajapakse et al., 2008). Brás et al. [59]".

Response 2: The comment was considered. The error in the sentence was rectified. See lines 72-73, 74-75, and 582.

  1. and 4. Line 135-145. What is the composition of pigment extract? Have you identified the active ingredients of the extract that showed promising biological activity? I think more experimental work is need. What is the chemical nature/composition/structure of the three extracts?

Response 3 and 4: The authors appreciate the reviewer's comment. However, Unfortunately, at this time, due to technical problems, the identification of the compounds by HPLC is not available. Nevertheless, in a previous publication, the chlorophylls, and carotenoids of the three pigment extracts were optimized and quantified ( https://www.sciencedirect.com/science/article/pii/S1319562X20306410 )

Reviewer 2 Report

·         This is a comprehensive study done to assess the blood-group dependency of antihemolytic activity as well as other activities of various extracts of Navicula incerta. Quantitative data are complemented by microscopic data, although the authors should take caution in interpreting results. The manuscript requires language edition and reorganization (particularly separating Results and Discussion).

·         In the toxicity bioassay, increasing concentration is not accompanied by decreased live cells, particularly with acetone extract at 250 ug/ml, as it is claimed to be the most active extract. The authors need to explain this inconsistency.

·         The use of letters (upper and lower case) in differentiating type of analysis performed as well as significant differences observed is quite confusing to the reader. The authors are advised to use less complex options. Maybe, the authors can present the data in Tables representing the within group (by asterix) and between group (Pound key or hush) analysis. As the ABO group has more than two groups the use of t-test can provide a false positive result. Hence, use the t-test for RH group (between group analysis). What’s important here is which blood group is susceptible to a given extract. Thus, results should be narrated first showing the blood group susceptible to the extract and this can be followed by which extract is more active.

·         Table 3: the presence of association is not dictated by the odds ration rather by the p-value and this can be seen easily from the 95% CI. If the interval excludes 1, then it is said to be significantly associated. The authors cannot arbitrarily designate OR <1 or >1 to define association.    

Author Response

Thanks to the reviewers for their comments.

  1. This is a comprehensive study done to assess the blood-group dependency of antihemolytic activity as well as other activities of various extracts ofNavicula incerta. Quantitative data are complemented by microscopic data, although the authors should take caution in interpreting results. The manuscript requires language edition and reorganization (particularly separating Results and Discussion).

Response 1: The authors appreciate the reviewer's comment. However, consulting with most of the authors, it was decided to keep results and discussions together. As support for a better understanding of the Results and Discussions, a better reorganization of each section and an exhaustive revision were carried out. The document was reviewed by an expert in the English language.

  1. In the toxicity bioassay, increasing concentration is not accompanied by decreased live cells, particularly with acetone extract at 250 ug/ml, as it is claimed to be the most active extract. The authors need to explain this inconsistency.

Response 2: The comment was considered. The authors explain this inconsistency. See lines 326-337.

  1. The use of letters (upper and lower case) in differentiating type of analysis performed as well as significant differences observed is quite confusing to the reader. The authors are advised to use less complex options. Maybe, the authors can present the data in Tables representing the within group (by asterix) and between group (Pound key or hush) analysis. As the ABO group has more than two groups the use of t-test can provide a false positive result. Hence, use the t-test for RH group (between group analysis). What’s important here is which blood group is susceptible to a given extract. Thus, results should be narrated first showing the blood group susceptible to the extract and this can be followed by which extract is more active.

Response 3: The comment was considered. The results are presented in the form of tables to improve the understanding of the data. The inconsistency of the t-test was corrected, and it was applied only for the Rh group. The authors considered comments on the susceptibility of the extracts and their activity. See lines 348-353, 356, 365-371, 395, 405, 416, 426-427, 441, 449-465, and 472.

  1. Table 3: the presence of association is not dictated by the odds ration rather by the p-value and this can be seen easily from the 95% CI. If the interval excludes 1, then it is said to be significantly associated. The authors cannot arbitrarily designate OR <1 or >1 to define association. 

Response 4: The comment was considered. The authors considered this inconsistency.  See lines 449-465, Now Table 3 is Table 7. See line 467.

Round 2

Reviewer 1 Report

Manuscript is ok now.

Author Response

Comment 1. Manuscript is ok now.

Response 1. Thank you.

Reviewer 2 Report

The authors addressed most of my comments. However, they need to consult a biostatistician as regards to the association study. 

Author Response

Comment 1. The authors addressed most of my comments. However, they need to consult a biostatistician as regards to the association study. 

Response 1. We appreciate the reviewer's comment. The statistical part was reviewed by an expert from our university and everything seems to be correct. Perhaps the reviewer can tell us specifically where it causes confusion, we could attend to it promptly.